# DEBAGREEMENT: A comment-reply dataset for (dis)agreement detection in online debates

**John Pougué-Biyong**[*]
University of Oxford
john.pougue-biyong@maths.ox.ac.uk

**Valentina Semenova**[*]
University of Oxford
valentina.semenova@maths.ox.ac.uk

**Alexandre Matton**
Scale AI
alexandre.matton@scale.com

**Rachel Han**
Scale AI
rachel.han@scale.com

**Aerin Kim**
Scale AI
aerin.kim@scale.com

**Renaud Lambiotte**
University of Oxford
renaud.lambiotte@maths.ox.ac.uk

**J. Doyne Farmer**
University of Oxford
doyne.farmer@inet.ox.ac.uk

## Abstract

In this paper, we introduce DEBAGREEMENT, a dataset of 42,894 comment-reply pairs from the popular discussion website `reddit`, annotated with *agree*, *neutral* or *disagree* labels. We collect data from five forums on `reddit`: `r/BlackLivesMatter`, `r/Brexit`, `r/climate`, `r/democrats`, `r/Republican`. For each forum, we select comment pairs such that they form altogether a user interaction graph. DEBAGREEMENT presents a challenge for Natural Language Processing (NLP) systems, as it contains slang, sarcasm and topic-specific jokes, often present in online exchanges. We evaluate the performance of state-of-the-art language models on a (dis)agreement detection task, and investigate the use of contextual information available (graph, authorship, and temporal information). Since recent research has shown that context, such as social context or knowledge graph information, enables language models to better perform on downstream NLP tasks, DEBAGREEMENT provides novel opportunities for combining graph-based and text-based machine learning techniques to detect (dis)agreements online.

## 1 Introduction

Online debates have a considerable impact on society. With over 4.2 billion people actively using social media, gaining insights into the evolution of online discussions and movements is important for explaining social change.[2] Fortunately, the field of Natural Language Processing (NLP) offers techniques to understand textual interactions. Specifically, one important research area is *(dis)agreement detection*, as it is fundamental for understanding societal polarisation and the spread of ideas online [31, 42, 34, 32].

(Dis)agreement detection falls under the field of *stance detection* [39] – the automatic classification of the position (or stance) of the producer of a piece of text, towards a target, into one of three classes:

---

[*]Equal contribution
[2]https://datareportal.com/reports/digital-2021-global-overview-report

35th Conference on Neural Information Processing Systems (NeurIPS 2021) Track on Datasets and Benchmarks.

*in favor, against*, or *neutral*. Due to the explosion of available online data sources, there has been a plethora of automatic natural language systems aimed at detecting stances. They have used either feature-based machine learning, deep learning, or ensemble learning approaches. A comprehensive review of these systems is presented in [23], section 5. One noticeable trend is the adaptation of pre-trained language representation models for stance detection, such as BERT [16], RoBERTa [27], DeBERTa [21] and XLNet [44], since they have led to considerable performance improvements for NLP tasks.

Although researchers have initially modelled stances using only text, recent work has shown that stance detection would benefit from context-sensitive approaches. In particular, several methods have leveraged graph (or network) features, such as interaction networks, preference networks, and connection networks. [12, 13, 6] use retweet data, while [33, 17] leverage hashtags to infer Twitter users' stances. Researchers have also explored how to incorporate social context [22, 2] and structured knowledge [10] into language models to improve inference on NLP tasks.

Despite the growing literature leveraging contextual features for stance detection, most existing datasets provide only textual information. The few datasets which provide contextual information are tweet datasets. These suffer from several drawbacks: i) they are shared via tweet identifiers, making it impossible to retrieve deleted tweet content and network information of deleted users for future research, and ii) retweets and hashtags may ease stance detection, but are features specific to Twitter discussions.

**Contributions**  We introduce DEBAGREEMENT, a dataset for detecting (dis)agreements in real-world online discussions. The dataset contains 42,894 comment-reply pairs, as well as contextual information (authorship, post, timestamp, etc), extracted from `reddit`. `Reddit`[3] is a social news aggregation, web content rating, and discussion website. This dataset presents opportunities to detect (dis)agreements by leveraging context beyond text and does not rely on platform-specific features, such as retweets or hashtags. Unlike existing datasets for stance detection, DEBAGREEMENT provides realistic online discussions, with diverse writing styles, genres and topics of discussion. We evaluate state-of-the-art (SOTA) pre-trained Language Models (LMs) on DEBAGREEMENT: our findings highlight ways to improve LMs with contextual information, and emphasize the substantial difference between DEBAGREEMENT and existing datasets. DEBAGREEMENT is available to download at: `https://scale.com/open-datasets/oxford`. Details of what data is contained within the dataset are available in the supplementary material.

**Impact**  DEBAGREEMENT presents new opportunities for modeling diverse online interactions with text and context (authorship, graph, temporal information). `Reddit`'s popularity, and the fact that all `reddit` discussions are downloadable for research purposes, make this a valuable data source to invest resources into understanding. Furthermore, the graph structure provided by DEBAGREEMENT offers opportunities for combining text-based machine learning (ML) and graph representation learning (GRL) methods. Modeling online discussion forums as graphs of interactions between users enables researchers to: i) translate the (dis)agreement detection task into a sign link prediction one [45], ii) use recent advances in GRL methods [20], and iii) leverage existing signed graph embedding methods [41, 15, 30] tested on publicly available signed graphs, such as Epinions and Slashdot [26, 25, 35].

DEBAGREEMENT follows active `reddit` users over time across five forums: `r/BlackLivesMatter`, `r/Brexit`, `r/climate`, `r/democrats`, `r/Republican`. It provides the possibility to investigate social theories, understand polarisation, and study how people express their opinions and change their views on social media.

## 2   Related datasets

[23] present a survey of recent advancements in stance detection and discuss 24 publicly available text datasets for stance detection (summarized in Table 6 of the survey). Among them, only two have more than 7,000 annotations [11, 14] and two are annotated for (dis)agreement detection specifically [4, 14]. The currently available datasets, summarised in [23], have several drawbacks: i) they only provide textual data, ii) all but two have fewer than 7,000 annotations, iii) they span a small range of

---

[3]`reddit.com`: the 20[th] most visited site globally as of March 2020

topics and text genres, and iv) their genre and writing style of the text is often formal and structured. The latter point often simplifies the task of stance detection, however, is not representative of online debates on popular platforms. For instance, in the popular Perspectrum dataset [8], claims (e.g. *Animals should not be used for scientific or commercial testing*) and perspectives (e.g. *Any living entity should not be treated as objects or property as doing so allows them to be treated amorally*) are presented in a formal writing style.

In the literature, only two publicly available datasets contain contextual data along with the text and are annotated for stance detection in social media . [36] provide 11,398 annotated Spanish tweets related to the 2017 Catalan Referendum (in favor of the independence, against, or neither), as well the images and tweets before and after these tweets on users' timelines. [9] present 3,282 annotated Italian tweets related to the 2019 sardines movement. The authors provide contextual information based on the tweet (number of retweets, likes, replies and quotes received to the tweet, type of posting device, date), the tweet's author (number of followers, of tweets ever posted, user's bio) and their social network (friends, replies, retweets, quotes' relations). However, as pointed out by [23, 1], these datasets suffer from the following drawbacks: i) tweet datasets are shared via tweet identifiers, making it impossible to retrieve deleted tweet content and network information of deleted users for future research, and ii) retweets and hashtags may ease stance detection, but are features specific to Twitter discussions. Furthermore, Twitter does not provide complete, open-source data access for research purposes.

## 3 Debagreement

### 3.1 Collection

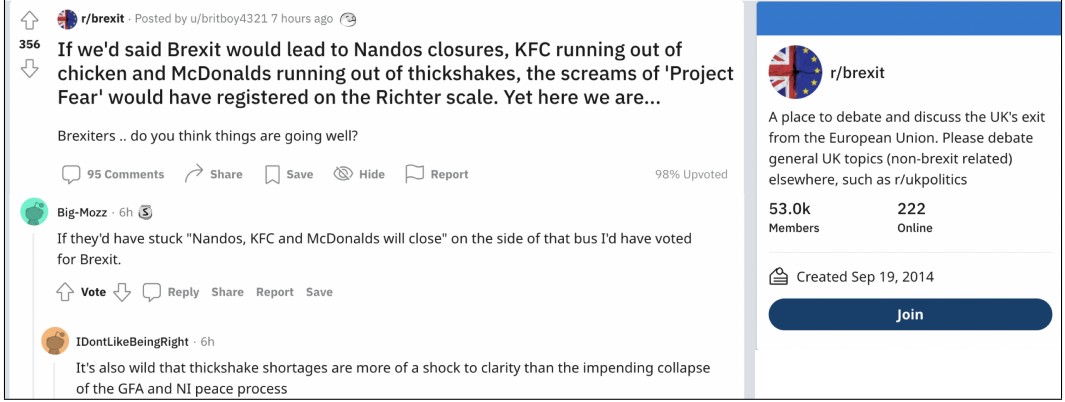

Figure 1: `r/Brexit`

DEBAGREEMENT was created by retrieving data from a popular discussion forum website, `reddit`. Reddit is broken down into discussion sub-forums, called subreddits, and denoted `r/*` (e.g. r/Brexit). Within each subreddit, users write titled posts, typically accompanied with a body of text and/or a link to an external website. These posts (or submissions) can be commented, *upvoted* or *downvoted* by other users (Figure 1). A ranking algorithm raises the visibility of a submission based on the number of upvotes it receives, but lowers it with time, so that the first posts visitors see are highly upvoted and/or recent. Comments related to a post are also visible, and are subject to a similar scoring and commenting system.

We collected `reddit` data from the PushShift API.[4] The API provides historical `reddit` data, which is open-source for researchers [5]. We pull data from three subreddits of social movements `r/Brexit`, `r/BlackLivesMatter`, `r/climate`, and two subreddits of political affiliations `r/Republican`, `r/democrats`. A description of each subreddit, summarised from the subreddit websites, along with usage counts (as of August 2021), are presented below:

- `r/BlackLivesMatter` discusses news related to the *Black Lives Matter* movement. It was created in 2014 and has 109K members,

---

[4] `https://pushshift.io/`

- `r/Brexit` aims to foster debate about the United Kingdom's (UK) exit from the European Union (EU). It was created in 2014 and has 53K members,
- `r/climate` is a community for truthful science-based news about climate and related politics and activism. It was created in 2008 and has 99K members,
- `r/democrats` is a partisan subreddit. It aims to discuss political news, policies and how to ensure the election of Democratic party candidates. It was created in 2014 and has 292K members,
- `r/Republican` is a partisan subreddit for Republicans to discuss issues with each other. It was created in 2008 and has 172K members.

Data from `reddit` is anonymous: users choose usernames which are not associated with any personally identifiable information. We further discuss ethics and anonymity in the supplementary material.

For `r/climate` and `r/Brexit`, we collect all the submissions and posts since the subreddits' creation dates until May 2021, in order to track these social movements from their inception. For `r/BlackLivesMatter`, `r/democrats` and `r/Republican`, we collect data from January 2020 in order to focus on recent, critical events: the protests following George Floyd's murder and the 2020 United States presidential election.

**Data cleaning** We first excluded empty comments, comments from deleted authors, comments that were hidden for user privacy reasons, and comments containing hyperlinks. Inside the text body of both the submissions and the comments, we removed paragraph breaks and replace a small set of special characters (for example, *&amp* is replaced with *and*). Despite potentially offensive content being present in the discussion forums, we purposefully do not filter out this content in order to realistically capture the nature of online discussions.

**High-quality interactions** A large amount of submissions receive no or few comments on `reddit`. Respectively 92%, 45%, 83%, 75%, and 59% of posts received less than 5 comments in `r/BlackLivesMatter`, `r/Brexit`, `r/climate`, `r/democrats`, and `r/Republican`. In order to annotate impactful discussions in a given subreddit, we remove posts with fewer than 10 words and with fewer than $k$ comments, where $k$ is the rounded average number of comments per submission on each forum ($k = 2, 5, 5, 5, 10$ for `r/BlackLivesMatter`, `r/climate`, `r/Republican`, `r/democrats`, and `r/Brexit`, respectively). We also filtered out comments with fewer than 10 or more than 100 words, and comments that contain hyperlinks. We truncate submissions to 100 words in length, as this is sufficient to contextualize the interaction for annotators. After further inspecting the nested discussion structure on `reddit`, it became apparent that discussion threads between users with different opinions often devolve into an onslaught of negative affronts with little substance. Conversely, conversations between people with similar opinions tend to contain only a few nested comments. These observations motivated our decision to retain only comment-reply pairs whose parent comment replied directly to the initial submission (nest level 1), as proxies of lengthier discussion threads between two given users. We also remove comments whose authors have been deleted or whose contents have been removed. The percentage of data affected by each filtering step is provided in the supplementary material.

**Graph creation** For each subreddit `r/*`, the resulting set of interactions forms a multi-edge, temporal graph $\mathcal{G}_{r/*}$, where nodes are users, and edges represent a comment-reply interaction between two users. One of the unique advantages of DEBAGREEMENT over other datasets is the additional graph interaction information provided about every subreddit.

For all of the forums except `r/Brexit`, we keep all comment-reply pairs as the final dataset. However, because $\mathcal{G}_{r/Brexit}$ had significantly more comment-reply pairs than the other forums, we retained only the users (nodes) who commented on at least ten posts over the course of a given month. The number of nodes and edges in each graph is provided in Table 1. The final dataset is comprised of a total of 49,140 comment-reply pairs, forming a temporal user interaction graph for each subreddit.

## 3.2 Annotation

**Crowdsourcing annotation setup** Comment-reply pairs were annotated with *agree*, *neutral*, *disagree*, or *unsure* labels by a team of 529 English-proficient annotators from Scale AI. The annotators

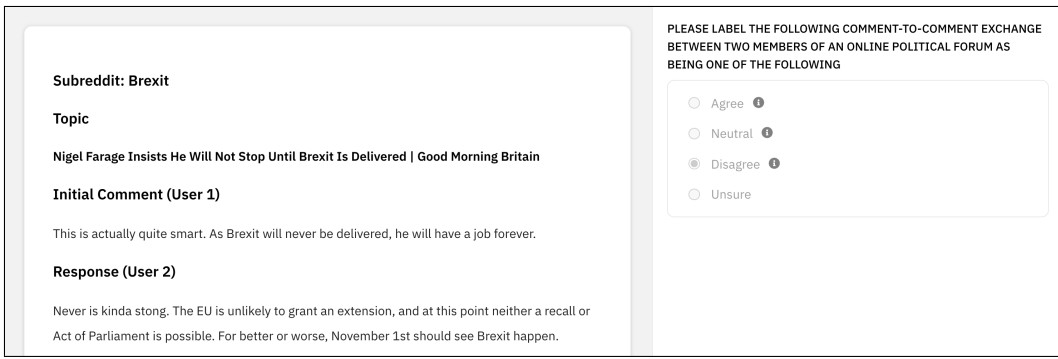

Figure 2: User interface for annotators

were provided with introductory courses on the BLM and climate change movements, as well as UK and US politics. They were also provided with examples for each of the four labels, with the reasoning behind the labels attributed.

Annotators were trained using a combination of i) an instruction document, ii) training webinars taught by Scale AI's internal subjective matter expert, and iii) training courses and quizzes based on the instructions and a sample golden dataset. The final set of annotators was selected based on their performance on a gold standard dataset of 74 comment-reply pairs labelled by the authors of this paper.

Each annotator was given a confidence score based on their accuracy on the gold standard dataset. Annotators were presented with a ten page 'Instructions' document, accompanying each task, as a reminder of annotation best-practices and with ten examples. Details from the instructions document are presented in the supplementary material. The annotator user interface is depicted in Figure 2: each task includes the subreddit name, the initial post (topic), the comment-reply pair, and a list of subreddit-specific abbreviations. Annotators are being compensated above minimum wage in their respective labor markets in order to ensure ethical and fair compensation.

**Labelling and inter-annotator agreement**   Each task is annotated by three to five annotators based on Scale AI's dynamic consensus process. Specifically, the number of annotators per task is determined by averaging the confidence score of the annotators and comparing it against a minimal desired confidence threshold score. For each task, the majority class was decided as the final label. Tasks where the annotators equally chose *agree*, *neutral* and *disagree* were reviewed by Scale AI's internal subject experts before finalizing the response. Overall, 33% of the annotations have full inter-annotator agreement. We drop the 6,246 comment-reply pairs whose final label is unsure, and the pairs with lower than 2/3 inter-annotator agreement score, leaving a final dataset of 42,894 labeled interactions. The statistics of DEBAGREEMENT are detailed in Table 1.

Table 1: Dataset statistics

|  | r/Brexit | r/climate | r/BLM | r/Republican | r/democrats |
|---|---|---|---|---|---|
| Start date | Jun 2016 | Jan 2015 | Jan 2020 | Jan 2020 | Jan 2020 |
| #nodes | 722 | 4,580 | 2,516 | 8,832 | 6,925 |
| #edges | 15,745 | 5,773 | 1,929 | 9,823 | 9,624 |
| *positive* | 29% | 32% | 45% | 34% | 42% |
| *neutral* | 29% | 28% | 22% | 25% | 22% |
| *negative* | 42% | 40% | 33% | 41% | 36% |

## 3.3   Analysis

**Activity**   Figure 3 displays how the number of interactions in DEBAGREEMENT varies over time. The activity matches key historical events in several subreddits. For example, the activity in r/Brexit grew dramatically as the original date of the UK's exit from the EU approached. On the

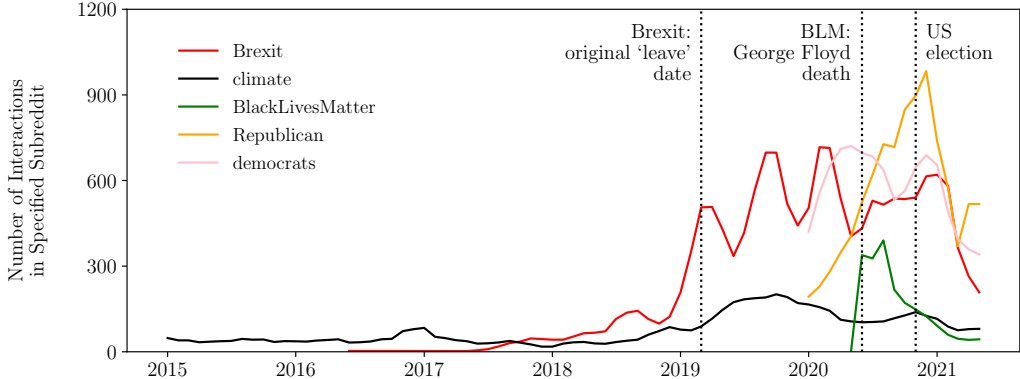

Figure 3: Number of interactions per subreddit (3-month rolling averages)

other hand, `r/BlackLivesMatter` rose to prominence following the death of George Floyd. We also observe activity spikes in `r/democrats` and `r/Republican` following the 2020 United States presidential election.

**Disagreements and polarisation**  Increased research efforts have been made to understand polarisation in social media [38, 3]. In addition to offering a valuable dataset for the NLP community, DEBAGREEMENT also presents opportunities to study online social movements. This paragraph gives an overview of the insights that can be gained by using the DEBAGREEMENT dataset by taking a closer look at the annotated (dis)agreement data from `r/Brexit`.

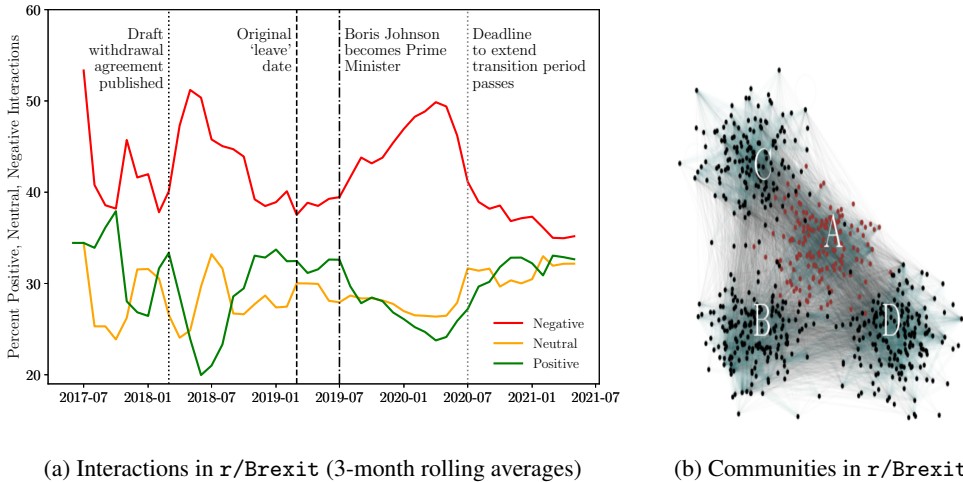

(a) Interactions in `r/Brexit` (3-month rolling averages)  (b) Communities in `r/Brexit`

Figure 4: Polarisation in `r/Brexit` over time

Figure 4a displays how positive, negative and neutral interactions have evolved in `r/Brexit` over time. We observe that the publication of the initial draft withdrawal agreement brought division within the forum, with a drastic drop in the fraction of agreements and a rise in disagreements. Similarly, the election of Boris Johnson as Prime Minister and the subsequent negotiations were both correlated with a gradual rise in disagreements.

When aggregating the data over the whole time period in `r/Brexit`, the resulting (static) graph of discussions offers additional insight into polarisation. We mine communities of individuals with similar opinions by applying a conventional community detection algorithm for signed graphs [37] to the annotated `r/Brexit` data. Such algorithms aim to find graph communities by maximising the number of positive edges within communities and the number of negative edges between communities. As depicted in Figure 4b, we obtain four communities. By reading the comments posted by the

most active users of each community, we conclude that community **A** (in brown) is pro-Brexit and communities **B**, **C** and **D** (in black) express sentiments in favor of the UK remaining in the EU. We look further at the main topics of discussion in each community and conclude that: users in community **B** are interested in the consequences of Brexit on international trade, users in community **C** discuss the accountability of UK political figures in what they consider 'a disaster' for the UK, and users in community **D** are mostly interested in UK-EU negotiations and the votes in UK parliament.

DEBAGREEMENT is available to download at: `https://scale.com/open-datasets/oxford`.

# 4 Benchmark evaluations

## 4.1 Experimental Setup

All experiments were performed on two NVIDIA TITAN RTX 24Gb GPUs. We use the `HuggingFace` implementation of the language models we evaluate.

## 4.2 Evaluating SOTA pre-trained LMs

Table 2: LMs accuracy for (dis)agreement detection: standard deviation in parentheses

| BERT | | | XLNet | | RoBERTa | | DeBERTa | |
|---|---|---|---|---|---|---|---|---|
| base/uncased | base/cased | large/uncased | base | large | base | large | base | large |
| 62.4 % (0.6%) | 61.8% (0.1%) | 63.7% (1.2%) | 63.6% (0.8%) | 63.3% (0.6%) | 63.2% (1.4%) | 64.1% (1.3%) | 63.0% (0.6%) | **64.1%** **(0.8%)** |

We evaluate the performance of four state-of-the-art pretrained language models on (dis)agreement detection: BERT [16], RoBERTa [27], DeBERTa [21] and XLNet [44]. We build training samples by concatenating the parent sequence, the $[SEP]$ token, and the child sequence. We split the data into 80%/10%/10% train/val/test sets while maintaining the temporal order, where testing is done on the latest data. This follows a realistic setting in which one uses a trained model to perform prediction on new, incoming data [24]. The chosen train/val/test split prevents data leakage of terminology only used in future periods into training data. We train each model four times with different seeds. All models perform with average accuracy ranging from 62% to 64% (+22/24% above the majority class).

**Failure modes** Due to relatively similar model performances, we choose to focus on BERT base/uncased (b/u) to gain further insight into LM failure modes.

Table 3: BERT(b/u) performance statistics

| | precision | recall | f1-score | support |
|---|---|---|---|---|
| disagree | 64.3% | 73.8% | 68.7% | 557 |
| neutral | 63.7% | 44.1% | 52.1% | 517 |
| agree | 60.1% | 69.4% | 64.4% | 500 |

Table 4: BERT(b/u) confusion matrix

| | | Predicted Label | |
|---|---|---|---|
| | **-** | **0** | **+** |
| **-** | 74% | 11% | 15% |
| **0** | 28% | 44% | 28% |
| **+** | 17% | 14% | 69% |

(True Label on vertical axis)

The model performs worst at identifying *neutral* interactions. Qualitatively, examples of *agree* and *disagree* classes show clear support or animosity between two users. On the other hand, *neutral* examples, which lack engagement and/or express partial agreement and disagreement, have the potential to confuse the model.

We present a comment-reply pair which BERT(b/u) labeled as *agreement* with high confidence, however, annotators correctly identified as a *neutral* interaction:

*Subreddit*: Brexit

*Parent*: The government don't have anything against immigrants personally. In fact they know that the economy thrives on healthy immigration. But they capitalized on a xenophobic voter base, so they have to sneak stuff like this in under the radar.

*Child*: Although I think you're right the government knows the UK needs immigrants I'm not so sure about their personal opinions. Mrs. May did a lot to make life harder for immigrants with her hostile environment policies. Much more than was necessary to appease the xenophobic voter base.

---

We observed that interactions with the largest loss values are often labeled *agree* or *disagree* by BERT(b/u), even though they are considered *neutral* by annotators. This points to the fact that current SOTA LMs still fail to capture the subtleties of human dialogue when complex or nuanced interactions occur [18, 29].

**Formal vs. informal online interactions**   We argue that: i) current LMs struggle with messy data when it is either scraped directly from social media or pulled from human dialogue, and ii) clean and formal datasets may not have transferable insights for online (dis)agreement detection.

As outlined in section 2, most existing stance detection datasets are either small, focused on one particular topic, and/or contain formal, structured discussions. We consider the Perspectrum dataset [8], a formal text dataset for (dis)agreement detection, and compare BERT(b/u) performance on DE-BAGREEMENT and Perspectrum. For consistency, we retain only *support/oppose* annotations in Perspectrum, and *agree/disagree* interactions in DEBAGREEMENT.

Table 5: Accuracy of BERT(b/u) - DEBAGREEMENT vs Perspectrum

|  | Brexit | Republicans | Democrats | Climate | BLM | Perspectrum |
|---|---|---|---|---|---|---|
| All subreddits | 82.1% | 78.4% | 81.0% | 83.9% | 79.2% | 57.7% |
| Perspectrum | 56.4% | 54.1% | 57.1% | 55.2% | 58.4% | 90.5% |
| Most frequent class | 52.5% | 52.7% | 51.0% | 58.3% | 69.8% | 52.9% |

In Table 5, we compare the accuracy of BERT(b/u) when trained on all subreddits (first row) or Perspectrum (second row), and tested on each subreddit and Perspectrum (columns). We observe an accuracy of 90.5% when training on Perspectrum and evaluating performance on the same dataset. However, BERT(b/u) trained on Perspectrum performs poorly on DEBAGREEMENT. The fine-tuned model even fails to outperform the naive most-frequent class estimate for subreddits where the class balance is poor (`r/BlackLivesMatter` and `r/climate`). These findings imply that disagreement detection in the online, messy setting is a fundamentally different problem to the formalised, structured setting, with relatively few transferable insights between them.

## 4.3   Alternative training data

In this section, we consider the performance of BERT(b/u) with different choices of training data from DEBAGREEMENT. We show that i) BERT(b/u) performs better on a specific subreddit when trained on data from other subreddits, and ii) masking either of the two comments during training leads to lower performance.

**Cross-subreddit evaluation**   We consider BERT(b/u) performance on a specific subreddit when trained on the subreddit, other subreddits, or all subreddits. In Table 6, we observe that training on data across different subreddits improves performance compared to training on the test subreddit alone. This suggests the capacity of LMs to learn signal beyond subreddit-specific terminology and jargon.

**Masking parent and child comments**   We also assess the relative importance of the parent and child comments for the (dis)agreement detection task. We compared the accuracy of BERT(b/u) trained with both the parent and child comments, and trained with each one of them only, on `r/Brexit`. The baseline accuracy (when the model is trained with both comments) is 62.4%. We observed an accuracy of 38.1% on the test set when we trained the model with the parent message only, and an accuracy of 60.7% when training with the child message only. This suggests that: i) the

Table 6: Cross subreddit BERT(b/u) accuracy

| | | Brexit | Republicans | Democrats | Climate | BLM |
|---|---|---|---|---|---|---|
| | | | | Subreddit | | |
| Training data | Most frequent class | 35.4% | 40.3% | 39.7% | 41.2% | 53.6% |
| | Current subreddit | 62.4% | 64.6% | 65.3% | 62.2% | 58.3% |
| | Other subreddits | 62.2% | 64.3% | **66.9%** | 65.4% | **70.8%** |
| | All subreddits | **64.1%** | **64.9%** | 66.9% | **66.1 %** | 68.6% |

child comment contains most of the signal required for the task, and, most importantly, ii) the parent comment provides textual context which improves (dis)agreement prediction.

## 5 Limitations and Future work

**Combining LMs and GRL methods**   Recent research has proven the benefits of leveraging contextual information in language modeling, whether it is structured knowledge [10] or social context [22, 2]. GRL methods have become increasingly popular for natural language processing [43]. [10] reviews how structured knowledge, such as knowledge graph (KG) embeddings [40], have been combined with language models. An example is KnowBERT [28], a language model which injects KG embeddings into BERT's internal layers so the model learns knowledge-aware token representations. In another context, [2] build a geographically-sensitive language model for token prediction in tweets of the form: *The most popular NFL team in my state is [MASK]*. The authors use Node2Vec [19] on a graph in which nodes are US cities and edge weights are geodesic distances between cities. They show that injecting the embedding of the location of the tweet author into the internal layers of BERT improves BERT's performance.

DEBAGREEMENT presents an opportunity to train socially aware language models. For instance, one may use user embeddings trained on the DEBAGREEMENT graphs, and inject them into state-of-the art LMs. One may also consider (dis)agreement detection as a sign link prediction task on the DEBAGREEMENT graphs. In this setting, the textual information could be treated as edge or node features, and suitable Graph Neural Networks (GNN) may be applied for edge sign prediction. Social balance theories (such as the theory that *A friend of my enemy is my enemy*) may be relevant for inferring missing edges [7].

**Labeling challenges**   Despite the preprocessing we performed to ensure high-quality annotations, only 33% of the pairs were annotated with full inter-annotator agreement. This speaks to the difficulty in annotating short online exchanges containing nuanced statements, sarcasm, and diverse writing genres. Furthermore, the boundary between a neutral and a negative reply, or a positive one, may be blurred. This is underscored by the difficulty LMs have to identify *neutral* interactions.

We suggest two potential ways to combat labeling challenges through future work. One involves drawing more data from the forum for better context - specifically, we suggest potentially annotating full discussion threads, instead of comment-to-comment interactions in isolation (see below). Additionally, we see some potential in training NLP tools on interactions where annotators did not agree to identify examples that need additional attention.

**Comment-reply pairs vs discussion threads**   As discussed in section 3.1, `reddit` users may engage in lengthy back-and-forth exchanges, rather than a single comment-reply interaction. In DEBAGREEMENT, we consider initial comment-reply pairs as a proxy for discussion threads. By doing so, we may fail to capture some of the complexity of an entire online conversation, and opportunities to foster dialogue modeling research.

**Exploring more subreddits**   DEBAGREEMENT follows five key socio-political movements. Expanding annotations to other areas of discussion (such as consumer products reviews, `r/buyitforlife`, wider political forums, `r/politics`, investor-related forums, `r/WallStreetBets`) could lead to further discoveries. Annotated data from these subreddits may allow us to glean new insights about future consumer preferences or retail investor decision-making.

# 6 Conclusion

In this paper, we introduce DEBAGREEMENT: a dataset of 42,894 comment-reply interactions annotated with (dis)agreement labels, with authorship, graph, topic and temporal information. The dataset offers several advantages over existing datasets, including its size, its contextual information and a less structured, but more realistic, language style. We perform several benchmark experiments on SOTA language models. The dataset presents promising research avenues for combining text with contextual information.

## Acknowledgments and Disclosure of Funding

We would like to thank Matteo Bruno, James Caulkins, Xiaowen Dong, Peter Grindrod, Jack Hessel, Sean Holloway, Dorothy Nicholas, Janet Pierrehumbert, Julian Winkler for their valuable help and fruitful discussions. We also want to thank our NeurIPS reviewers for great feedback on the paper.

We thank Baillie Gifford, the Institute for New Economic Thinking at the Oxford Martin School, and the UK Engineering and Physical Science Research Council for funding our work at the University of Oxford.

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
