# OpenReview forum: "DEBAGREEMENT: A comment-reply dataset for (dis)agreement detection in online debates"
_NeurIPS.cc/2021/Track/Datasets_and_Benchmarks/Round2 — NeurIPS 2021 Datasets and Benchmarks Track (Round 2)_

### Official Review · Reviewer_bb3G · 2021-09-15
**Valuable dataset, although some choices could be better motivated**

**Rating:** 7
**Confidence:** 4
**Clarity:** Overall the paper is well written.

**Strengths:**


* The dataset is large compared to existing datasets. The focus on Reddit data complements existing datasets that are often based on Twitter data. I can see that this dataset will be useful to researchers interested in stance detection and online discussions.
* Overall the paper is well written.
* A nice set of experiments, such as investigating the effect of training/testing on different subreddits and the importance of context (e.g. parent post) for prediction (vs. the child message).


**Weaknesses:**

- Some choices in the preprocessing/filtering/cleaning process are not clearly motivated
- More details on the annotation would be useful (inter-annotator agreement, analysis of cases with low agreement)
- More examples of the data in the main paper would be useful
- Paper doesn’t show empirically that the contextual information (e.g. social graph) is important for this task.


**Additional Feedback:**

* What do you mean with "social evolution"? (line 18)
* "to understand" what? (l56)
* Table 2: What are the numbers in parentheses?
* Appendix: line 649 [Insert website address].
* The section on combining LMs and GRL methods seems odd at its current position. The rest of the section presents actual results, and this is what I expected when I started reading this paragraph. I would move this to a different section in the paper (e.g. future work).
* Appendix: "minimally altered online interactions" what kind of changes were made? Are these the contractions that were expanded?
* the way Reddit is formatted in the text isn't consistent.
* I understand that space is limited, but more examples of the data in the main paper would have been very helpful, especially since the only example that was provided wasn't very clear (one could argue for a different label).

**Correctness:**

Overall the experiments are carefully set up, e.g. reporting results over different random splits and a train/dev/test split according to time. I also appreciate the set of experiments, such as cross-subreddit evaluation and the use of parent and/or child comments for classification.

* I wasn’t fully convinced by the argumentation in the section on ‘realistic online interactions’. For example, where do you show that LMs struggle with messy data when it is scraped directly from social media/pulled from human dialogue?
* Did you control for the size of training data when comparing performance of models trained on Perspectrum vs DEBAGREEMENT?

I had quite some questions/concerns about the choices made when constructing/cleaning the dataset:
* Why were comments with hyperlinks removed (125)? Isn’t it the case that users that engage in online discussions often refer to external sources (e.g. using urls) to back up their claims?
* I couldn’t follow the reasoning in lines 139-140. Why exactly did the authors choose to only focus on comment-reply pairs?
* For all the steps that were performed, it would be useful to report how many posts/comments were excluded because of each step. And how do these different filtering steps affect the sparsity of the interaction graph?
* The paper mentions that this dataset provides “diverse writing styles”, why then are contractions expanded?

Overall the annotation is carefully set up: annotators were provided with introductory courses on the relevant topics and a ten-page instructions document. However:

* The paper doesn’t clearly state the inter-annotator agreement (the paper mentions a few statistics, but doesn’t report inter-annotator agreement). Please specify it and be explicit about the measure used. What is meant with “full inter-annotator agreement”? (line 170)
* Line 170: Quite a large number of comment-pairs were dropped where the final label was unsure/low agreement. It would be important to better understand what types of posts these are. Actually, I think it can be valuable to not only provide the final label, but the full distribution of annotations when releasing the data, including the comment-reply pairs with low agreement. Assuming the annotators did their best, low agreement can be an interesting/valuable signal. I was also wondering if by filtering out low-agreement (‘difficult’) pairs, the performance on this task will be a bit inflated compared to using this system “in the wild”.

One of the main arguments of this paper is the value of the contextual information in their dataset, in particular the social graph. They don’t show this empirically e.g. through prediction experiments. I don’t consider this a reason to reject this paper, but the paper would have been more strong if they had actually shown this empirically.


**Documentation:**

It would be useful to provide
- the full set of instructions the annotators received (the ten page document)
- the set of 74 gold comment reply pairs.
- hourly wage of participants. It isn’t reported due to the industrial partner. Still, it would be uesful to say something about this even without the full details, e.g. the participants were paid above minimum wage… (assuming that this is the case)


**Ethics:**

No major concerns, but the discussion on the use of Reddit data could be more nuanced, both in the main paper and in the appendix.

Line 117: "*Data from Reddit is anonymous..*" I feel this is too strong. Users can have usernames
that may reveal personal information, or use the same username across different online platforms (which may make it easier to link/identify them). Similarly, users can still disclose personally identifiable information in posts.


**Relation To Prior Work:**

The paper clearly outlines what this dataset offers compared to already existing datasets. This is mostly based on a survey on stance detection from 2020. The paper could engage more with NLP papers related to argumentation, especially the ones looking at online debates, sometimes using Reddit data (e.g. see 'Winning Arguments: Interaction Dynamics and Persuasion Strategies in Good-faith Online Discussions' Tan et al. , ‘I Couldn’t Agree More: The Role of Conversational Structure in Agreement and Disagreement Detection in Online Discussions’ Rosenthal and McKeown, and others). What kind of data is used, and what kind of research questions have researchers been interested in?


**Summary And Contributions:**

This paper presents DEBAGREEMENT, a dataset with almost 43k comment-reply pairs from Reddit, annotated with agree, neutral and disagree labels. The dataset supports research on (dis)agreement detection. Furthermore, the released data can be used to construct a social graph, so that the incorporation of such contextual information in such detection models can be investigated. The paper presents several baseline experiments using state-of-the-art transformer models.

---

### Official Review · Reviewer_wqeX · 2021-09-20
**An useful dataset for dis/agreement detection task**

**Rating:** 9
**Confidence:** 4
**Correctness:** To my knowledge yes.
**Clarity:** This paper is very well written. Howe…

**Strengths:**

1. This dataset appears to be beneficial for dis/agreement detection task by providing fine-grained annotation and data targeting multiple domains.

2. The results are promising.

**Weaknesses:**

1. Only one existing dataset is used to make comparison with the dataset proposed, while the authors review several related datasets in section 2. It is also not clear why the dataset Perspectrum was selected for evaluation.

2. It is not clear how the the annotation instruction is given to annotators.

3. I am not so sure that it is the best strategy to discard pairs, which are quite a lot, labeled with "unsure" or has less than 2/3 inter-annotator agreement score. In fact, it reflects how difficult the task is, even with ten page instruction. Perhaps those pairs can be (partially) kept by running through a conciliation phase where annotators discuss and agree with a final label.

**Additional Feedback:**

N/A

**Documentation:**

I would expect to see the ten page Instructions document [line 157] for annotation? Is it available somewhere?

**Ethics:**

To my knowledge, no.

**Relation To Prior Work:**

It is well-motivated.

**Summary And Contributions:**

This paper introduce a new dataset for dis/agreement detection. The data was retrieved from Reddit covering 5 topics (BlackLivesMatter, Brexit, climate, democrats, Republican). In comparison with existing datasets, the proposed dataset is larger (in total 42, 894 pairs), contains contextual information of the pairs (e.g., authorships, post, timestamp) and annotation with one of 4 labels (agree, disagree, neutral, unsure), and provides realistic discussions as it is written by real users.

To establish benchmark evaluation, SOTA pretrained language models were finetuned on the dataset proposed and an existing dataset Perspectrum for detecting dis/agreement. The results appear to be promising, in particularly when considering contextual information.

---

### Official Review · Reviewer_ZxZa · 2021-09-23
**Valuable dataset and deep analyses**

**Rating:** 7
**Confidence:** 4
**Clarity:** This paper is well organized and easy…

**Strengths:**

1. This paper introduces a large-scale (dis)agreement detection dataset distinguished by the drawbacks of existing datasets. They provide detailed evaluations on the benchmark with experiments.
2. This paper proposes nice and deep analyses of the dataset in terms of both historical events and community groups (section 3.3).
3. The dataset containing contextual information in a graph format allows and suggests future work leveraging GRL methods which is one of the key ideas in research communities.

**Weaknesses:**

1. The limitations in Section 5, especially labeling challenges and comment reply pairs vs discussion threads, are plausible, but the authors do not provide feasible solutions or future works for those parts.
2. The authors should provide experimental results of combining LM and GRL methods compared to baseline LM models, if they aim consistently claiming that the graph format of the dataset is one of the strengths of this paper. Otherwise, this part should move into section 5 (future work).

**Additional Feedback:**

1. Minor typo) line 272: have been been combined -> have been combined
2. Comments) section 4.3, cross-subreddit evaluation: It shows opposite results compared to the recent works showing that stance detection would benefit from context-sensitive approaches (section 1, line 31). Could you please analyze why learning signals across different subreddits (contexts) improves the performances?
3. Question) section 4.3, masking parent and child comments: The paper claimed that training with the child message only shows 60.7% accuracy, while the baseline accuracy with full input is 62.4%. Does it imply any artifact or bias on the dataset?
4. Question) section 4.2, table 5: BERT trained with DEBAGREEMENT tends to fail on Perspectrum, while vice versa. Does it show that DEBAGREEMENT is not an improved version of (dis)agreement detection dataset but a different one?

**Correctness:**

This paper provides appropriate benchmark evaluations on state-of-the-art pre-trained LMs. Their experiments are constructed elaborately and performed with representative SOTA models.

**Documentation:**

There is documentation, “Dataset Terms of Use”, but it lacks sufficient details on availability/maintenance plan and intended uses. Also, it would be better to add a public URL if they have any plans on distribution. The authors provide sufficient details to get started with the dataset and reproduce the benchmark evaluations.

**Ethics:**

The authors should consider and mention whether both data collectors and data annotators are paid over living wage in terms of ethical issues.

**Relation To Prior Work:**

This paper points out the limitations of existing datasets and shows the comparison experiments between their dataset and the popular Perspectrum dataset.

**Summary And Contributions:**

This paper introduces DEBAGREEMENT, a comment-reply-based (dis)agreement detection dataset from controversial subreddits of the discussion website Reddit. DEBAGREEMENT overcomes several drawbacks of existing datasets, and the authors propose humanistic insights through inspections of the dataset. They provide benchmark evaluations on SOTA LM models and suggest leveraging GRL methods along with text-based methods.

---

### Official Review · Reviewer_eV85 · 2021-09-23
**A very well annotated (dis)agreement dataset is presented by the authors. The data is quite large and provides useful contextual information making the task more realistic.**

**Rating:** 8
**Confidence:** 4
**Correctness:** Evaluation methods and experiment des…
**Clarity:** Very Well written paper

**Strengths:**

Excellent dataset collected in a well defined process.


**Weaknesses:**

Some claims are not met in the paper. Specifically contextual graph Representation learning part.

**Additional Feedback:**

-

**Documentation:**

Instructions for the annotators should be made available.

**Ethics:**

Ananymity of users is not a given. Steps should be taken to ensure a utility preserving annotation of user names.

**Relation To Prior Work:**

Prior art discussed Sufficiently well.

**Summary And Contributions:**

A data with 42,894 comment-reply pairs from Reddit with authorship, graph, and temporal information. This is a very well-presented paper with a very useful dataset for stance identification.

Authors provide details of the collection and cleaning process which is very useful for authors for future related data collection efforts. High-quality content is retained with clear description of the process making it easily reproducible.

Crowdsourcing for data annotation is well defined (I would have liked to see the instructions). More details on IAA would have been welcome.

This is followed by an activity and polarisation analysis. While the analysis from these viewpoints is appreciated, it does not add much to the discourse of stance and agreement. The discussion of community graph is also illuminating but It is hard to see how (or if) these graphs were used by the authors in the experiments.

The argument on Real-world interactions is quite on point. More insights on how to resolve the issues would have been welcome.

Leveraging context is the primary pitch of the paper but while a significant amount of real-estate is devoted to GRLs it is not made clear if any experiments were attempted making use of these graphs (should have been a part of the.future work section)

---

### Decision · Program_Chairs · 2021-10-09

**Decision:**

Accept

**Comment:**

This is a well-annotated dataset of agreement and disagreement among Reddit users. The dataset would be very useful for agreement detection, and it is well constructed and well explained. It is also great that the social network can be built which could be of additional value. All reviewers agree that this paper should be accepted.